# Efficacy of Reciprocating Instruments in the Removal of Bioceramic and Epoxy Resin-Based Sealers: Micro-CT Analysis

**DOI:** 10.3390/ma14216670

**Published:** 2021-11-05

**Authors:** Marko Rajda, Ivana Miletić, Gorana Baršić, Silvana Jukić Krmek, Damir Šnjarić, Anja Baraba

**Affiliations:** 1Dental Health Center—Center, Runjaninova 4, 10000 Zagreb, Croatia; rajdamarko@gmail.com; 2Department of Endodontics and Restorative Dentistry, School of Dental Medicine, Gundulićeva 5, 10000 Zagreb, Croatia; miletic@sfzg.hr (I.M.); jukic@szg.hr (S.J.K.); 3Department for Quality, Department for Measurement and Control, Faculty of Mechanical Engineering and Naval Architecture, IvanaLučića 5, 10002 Zagreb, Croatia; gorana.barsic@fsb.hr; 4Department of Endodontics and Restorative Dentistry, School of Dental Medicine, University of Rijeka, 51000 Rijeka, Croatia; damirsnjaric@gmail.com

**Keywords:** bioceramic sealer, epoxy resin-based sealer, retreatment, micro-CT analysis

## Abstract

The objective of this study was to evaluate the efficacy of reciprocating instruments in the removal of bioceramic and epoxy resin-based sealers using micro-CT analysis. Root canals of 40 extracted human teeth were instrumented with a size R25 Reciproc instrument. Specimens were randomly divided into two groups (n = 20) according to the root canal filling material. In the first group, root canals were obturated with AH Plus sealer and Reciproc R25 gutta-percha. In the second group, a combination of bioceramic gutta-percha (TotalFill BC) and bioceramic sealer (TotalFill BC) was used. After one week, the retreatment of all specimens was performed using Reciproc instruments. To analyze the differences in the filling remnants, specimens were scanned in a micro-CT device after obturation and after the retreatment procedure. Statistical analysis was performed using the Mann–Whitney U test (*p* < 0.05). A combination of bioceramic sealer and bioceramic gutta-percha was more effectively removed from canals using a reciprocating instrument, with a filling remnants volume of 4.01 ± 3.13 mm^3^, in comparison to the combination of epoxy resin-based sealer and gutta-percha (6.96 ± 2.70 mm^3^) (*p* < 0.05). A reciprocating instrument was more effective in removing bioceramic sealers than epoxy resin-based sealers, although none of the root canal filling materials were completely removed from the root canals.

## 1. Introduction

The main goal of endodontic treatment is thorough chemo-mechanical debridement and removal of any inflamed, necrotic, or infected tissue so that root canals can be shaped and filled, thus ensuring a hermetic seal and preventing reinfection of the endodontic space [1]. According to the available literature, endodontic treatment is fairly predictable, and the success of primary endodontic treatment ranges from 86% to 98% [2]. However, in cases of failure or inadequate endodontic treatment, root canal retreatment is indicated [3]. The most common reasons for endodontic failure are poor control of aseptic conditions, missed canals, improper instrumentation of root canals, errors, and complications during root canal treatment in the form of perforations, separated instruments, short or overextended root canal filling, and microleakage of temporary or definitive restorations [3,4,5,6]. During root canal retreatment, root canal filling materials are removed from the root canals, followed by cleaning, shaping, and adequately obturating the endodontic space. According to one published review article, the success rate of endodontic retreatment is 78%, which is lower than the previously reported primary endodontic treatment success rate [7].However, a prospective study by [2] reported similar success rates for primary endodontic treatment (82.8%) and nonsurgical retreatment (80.1%) when the clinical procedures were performed by endodontic postgraduate students. Nevertheless, proper retreatment may be challenging, and complete removal of the existing root canal filling is not always possible.

A potential problem is that the remnants may operate as a mechanical barrier between the irrigating solution and the microbes that reside in hard-to-access areas, such as dentinal tubules, lateral canals, and isthmi, which might explain the lower success rate of the endodontic retreatment [1,6]. Furthermore, the residual material may adversely affect the adhesion of the new root canal filling to radicular dentin, which may also lead to failure [8]. The most commonly used material for root canal filling is gutta-percha, in conjunction with different root canal sealers. Depending on their chemical composition, sealers can be classified as zinc oxide eugenol, epoxy resin, silicon, calcium hydroxide, glass ionomer, methacrylate resin, or calcium-silicate-based sealers. The lattermost category includes bioceramic sealers that have been in use since their introduction in 2007 [9,10]. The chemical reaction of these sealers utilizes moisture in dental tubules, which then results in the formation of a calcium silicate hydrate gel and portlandite, while portlandite reacts with the phosphate ions in dental tubules and forms hydroxyapatite [11,12]. Furthermore, an experimental rat model study reported that two calcium silicate-based sealers showedthe potential to stimulate osteoblastic differentiation and promote the overexpression of osteo/cementogenic genes [13]. The histological evaluation by the von Kossa staining technique detected calcium precipitates in the fibrous capsule adjacent to bioceramicmaterial at 8 days; this isprobablydue to thealkalinity of the medium induced by the release of calcium ions, which would stimulate the formation of hydroxylapatite and the release of bone morphogenetic protein 2 and alkaline phosphatase, thus contributing to the mineralization process [13]. Although biomineralization enhances the adhesion of bioceramic material to root canal dentin, which is beneficial for successful sealing, it may also hinder the complete removal of the root canal filling if retreatment is needed. Retreatment may be time-consuming and sometimes requires substantial effort, but the use of engine-driven NiTi endodontic instruments allows for easier and faster instrumentation. Few studies have shown that single-file reciprocating instruments are rapid and effective in root canal retreatment [14,15]. Therefore, the present study aimed to evaluate the efficacy of reciprocating instruments in the removal of bioceramic and epoxy resin-based sealers using micro-CT analysis.

## 2. Materials and Methods

### 2.1. Sample Preparation

The study protocol was reviewed and approved by the Ethics Committee of the School of Dental Medicine, University of Zagreb, Croatia (05-PA-30-IX-9/2019). Forty (n = 40) single-rooted extracted human teeth with single canals were selected for the study. Teeth with root resorption, caries, root fractures, and/or previous endodontic treatment were excluded from the study. We removed all of the contaminated tissue outside the specimens before initial treatment and scanning, considering the aseptic conditions and safety of the operator. We used a hand scaler (iM3 Ergo Perio—Universal Scaler, Sydney, Australia).The crowns were sectioned below the cementoenamel junction using a water-cooled diamond drill. The working length was determined by inserting a size #15 K file into the canal until the tip of the file was at the apical foramen and then subtracting 1 mm. Root canals were instrumented using a size R25 VDW Silver Reciprocendomotor (VDW, Munich, Germany), according to the manufacturer’s instructions. During instrumentation, canals were irrigated with 2.5% NaOCl using a 27-gauge needle and a 2 mL syringe. To remove the smear layer, root canals were rinsed with 2 mL of 17% EDTA (pH = 7.7) for 1 min, followed by the final rinse with saline. The canals were dried using size R25 Reciproc paper points (VDW, Munich, Germany). All specimens were stored at 37 °C for one week to allow sufficient time for the sealer to set.

### 2.2. Obturation

The teeth were randomly divided into two groups (n = 20) based on the root canal filling material. In the first group (n = 20), root canals were obturated with a combination of AH Plus sealer (DeTreyDentsply, Konstanz, Germany) (Table 1) and size R25 Reciproc gutta-percha (VDW, Munich, Germany) using the single-cone technique. AH Plus sealer was mixed on a paper pad. Reciproc R25 gutta-percha was coated with AH Plus sealer, and then inserted up to the working length. In the second group (n = 20), a combination of bioceramic gutta-percha (TotalFill, FKG, La Chaux de Fonds, Switzerland, 25.06) and bioceramic sealer (TotalFill, FKG, La Chaux de Fonds, Switzerland) (Table 1) was used for obturation of root canals, again using the single-cone technique. TotalFill BC sealer was syringed into the canal, and TotalFill gutta-percha was placed in the canal up to the working length. After obturation, excess gutta-percha was removed with hot pluggers 1 mm from the cementoenamel junction. All specimens were stored at 37 °C for one week to allow sufficient time for the sealer to set. The canal access was sealed with glass ionomir cement (Equia Fil, GC, Tokyo, Japan).

### 2.3. Root Canal Retreatment

After one week and the complete setting of the sealer, root canal retreatment was performed for all specimens (n = 40) in both experimental groups. For retreatment, size R25 Reciproc instruments (VDW, Munich, Germany) with VDW were used. Silver Reciprocendomotors (VDW, Munich, Germany) were used according to the manufacturer’s instructions, and with no solvent. After three pecks, the root canals were rinsed with 2 mL of 2.5% NaOCl. The criteria for the completion of the retreatment procedure were smooth canal walls and no evident root canal filling material on the Reciproc files. At the end of the retreatment procedure, the root canals were rinsed with 2 mL of 17% EDTA (pH = 7.7) for 1 min, followed by the final rinse with saline. The canals were dried using size R25 Reciproc paper points (VDW, Munich, Germany).

### 2.4. Micro-CT Scanning and Analysis

The volume of gutta-percha was measured using a Nikon XT H 225 industrial computed tomography (XCT) machine at an X-ray energy of 110 kV and an X-ray tube current of 240 μA. These parameters were chosen to provide relatively fast scanning times while still providing a sufficiently small focal point size and therefore an adequate resolution. The X-ray source used in this experiment had a focal spot size–X-ray power relationship corresponding to approximately 1 μm per 1 W; in this case, 26.4 W of X-ray power resulted in a focal spot with a diameter of approximately 26 μm, which sets the fundamental hardware resolution limit prior to software subsampling. The final CT scan data had a voxel size of 36 μm, since 40 teeth were scanned together to ensure similar measurement conditions. Imaging was performed using a 400 mm × 300 mm 14-bit flat panel detector with 127 μm pixel size, exposed at 333 ms. CT data were acquired with 1440 projections with two frame-averaging algorithms. Scanned test plates were reconstructed using Volume GraphicsVGStudioMax.2 (v3.0, Volume Graphics GmbH, Heidelberg, Germany). Post-processing included beam hardening reduction using a Hanning filter (Volume Graphics GmbH, Heidelberg, Germany), noise reduction using a median filter, and surface detection using an adaptive search algorithm (Volume Graphics VGMax.2). During analysis, the filling material was treated as an inclusion in the base tooth material; this was possible because of the very distinct grayscale values of the tooth and filling material (10,000 and 40,000, respectively). The grayscale value of the tooth was used as the base material, and then a threshold algorithm was used to detect any occurrence of gutta-percha in the interior tooth volume. The results are expressed as a reduction of filling material volume on canal walls after the retreatment procedure (see Figure 1). After scanning, the images were reconstructed to 3D volumes using the CT Pro 3D software, Version XT 5.4 (Nikon Metrology Europe NV, Leuven, Belgium).

### 2.5. Statistical Analysis

Data were statistically analyzed using the SAS system software package (SPSS, v.20, IBM Corp., Armonk, NY, USA) for Windows platform. Statistical analysis was performed using the Mann–Whitney U test, and the level of statistical significance was set to α = 0.05.

## 3. Results

A combination of bioceramic sealer and bioceramic gutta-percha was more effectively removed from root canals using a reciprocating instrument, with a filling remnant volume of 4.01 ± 3.13 mm^3^, in comparison to a combination of epoxy resin-based sealer and gutta-percha (6.96 ± 2.70 mm^3^), (Table 2 and Table 3). The volume of the root canal filling materials decreased significantly in both experimental groups (*p* < 0.05), but none of the materials were removed completely (Table 3, Figure 2 and Figure 3).

## 4. Discussion

The present study tested the efficacy of reciprocating instruments in the removal of bioceramic and epoxy resin-based sealers using micro-CT analysis. The main aim of root canal retreatment is to eliminate microorganisms and their byproducts that sustain periapicalpathosis [16]. However, it is important to remove as much of the root canal filling material as possible to uncover areas with pulp tissue or bacteria that might have caused the failure of endodontic treatment [17]. By removing all the filling remnants, irrigating solutions or intracanal medicaments will be able to disinfect all areas of the endodontic space [18]. Several instruments may be used for the removal of gutta-percha and sealer, including hand instruments, burs, and engine-driven instruments, all of which are used with solvents or heat to soften the root canal filling material. A more recent engine-driven instrumentation technique that can also be used for retreatment employs a reciprocating motion. These instruments are used with a brushing motion, which might help in the removal of root canal filling materials [19]. According to one study [14], reciprocating instruments were more effective and faster in removing filling material than hand and rotary instruments [14]. Therefore, Reciproc instruments that employ reciprocating motion were chosen for instrumentation and root canal retreatment in the present study. However, to investigate the efficacy of only the reciprocating instrument in the removal of root canal fillings, no solvent was used during the retreatment procedure.

The efficacy of different techniques in removing root filling materials during retreatment can be assessed using different methods, and specimens are usually destroyed in the process. Previous studies have reported horizontal or vertical splitting of the examined teeth or clearing to render the teeth transparent [20]. The material remnants were evaluated by digital radiography [21], scanning electron microscopy [22], confocal microscopy [22], and optical microscopy [23,24,25]. Another method involves micro-CT analysis, which was also used in the present study. Micro-CT is a high-resolution imaging approach; it provides detailed imaging of the endodontic space and is based on accurate three-dimensional models that enable the acquisition of quantitative data without destroying specimens [26]. However, it cannot distinguish gutta-percha and sealer remnants [27].

Based on the results of the present study, none of the root canal filling materials were completely removed from the canal walls of any of the samples. These results are in accordance with the results of previous studies reporting that complete removal of filling material cannot be achieved by any retreatment method [28,29,30,31,32,33,34,35,36].

Furthermore, the combination of bioceramic sealer and bioceramic gutta-percha was more effectively removed from root canals using a reciprocating instrument than by a combination of epoxy resin-based sealer and gutta-percha. This result was somewhat surprising due to known tag-like structures composed of calcium and phosphate ions, which occur in the interaction of bioceramic material and root canal walls, and which are responsible for its sealing ability and dentine bonding [37,38]. One would expect that the bioceramic sealer would be more difficult to remove from the root canals. A possible explanation for these results can be found in a study in which confocal microscopy revealed a deeper penetration depth for epoxy sealer when compared to bioceramic sealer, which might explain the more effective removal of bioceramic sealer in the present study [22]. Other studies reported somewhat confusing data. According to some studies, epoxy and bioceramic sealers had comparable quantities of remaining materials [22,39]. Several studies obtained results according to which the removal of bioceramic sealer was less successful in comparison to epoxy resin [40], while several reported data in accordance with data fromthe present study [41]. Interestingly, studies [22,39,42,43] in which rotary endodontic instruments were used for retreatment did not show any difference in the removal of bioceramic and epoxy-based sealers. One possible explanation for these different results is that with single instrumentation using reciprocating techniques, such as the one used in the current study, the amount of debris removal seems to be reduced [44,45], which might also be true for the removal of existing root canal filling. Therefore, the use of multi-taper rotary instrumentation, such as that used in the aforementioned studies [39,40,41,42,43,44,45], might help in reducing bioceramic and epoxy-based sealerin a similar manner. Another explanation for these different results may be the different retreatment procedures, as well as the various methods used for the assessment of remnants of root canal filling materials in all cited studies.

## 5. Conclusions

The results of this in vitro study show that retreatment using reciprocating instruments is not always able to fully remove all the filling material from the canal; however, the group obturated with bioceramic sealers had less material remaining than epoxy resin-based sealers. We believe this information is important for clinical use, such as retreating teeth that have been previously obturated with bioceramic sealer.

## Figures and Tables

**Figure 1 materials-14-06670-f001:**
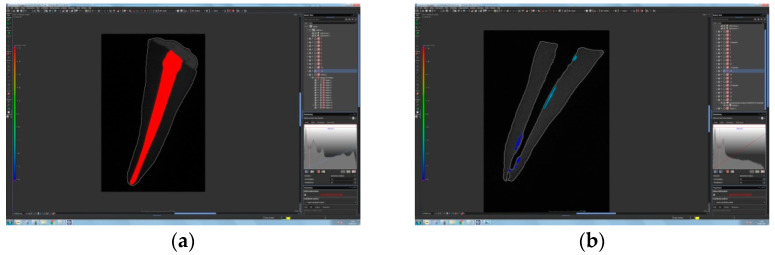
Cross-sectional micro-CT images of tooth specimens: (**a**) tooth specimen obturated with bioceramic sealer in combination with bioceramic gutta-percha; (**b**) same specimen after retreatment using Reciproc R25.

**Figure 2 materials-14-06670-f002:**
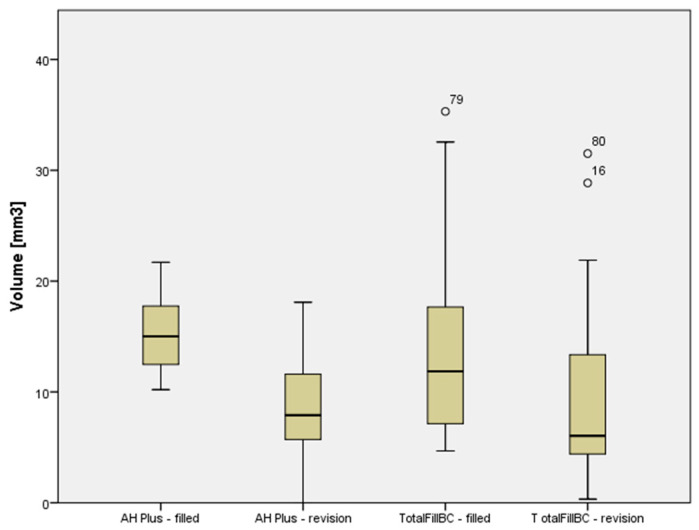
Plots of the scores given to the volumes of sealers in two groups before and after retreatment.

**Figure 3 materials-14-06670-f003:**
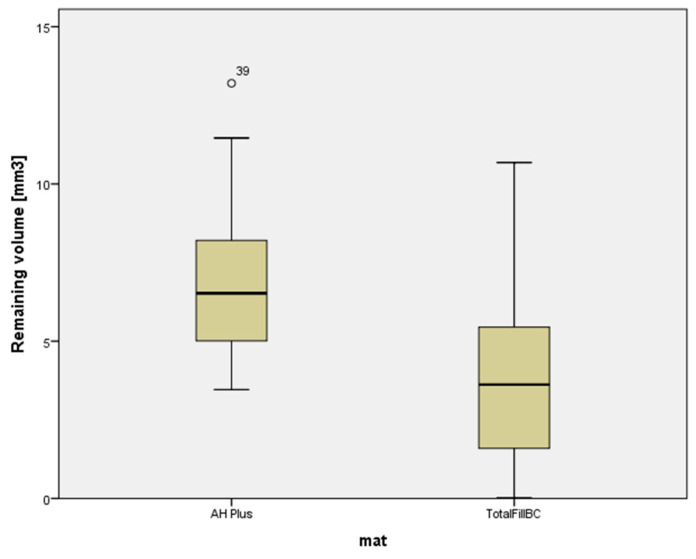
Plots of the scores dedicated to remaining filling material in two groups after retreatment.

**Table 1 materials-14-06670-t001:** Composition of root canal sealer agents.

Canal Sealer	Composition
AH Plus^®^	Paste A: bisphenol-A epoxy resin, bisphenol-F epoxy resin, calcium tungstate, zirconium oxide, silica, iron oxide pigmentsPaste B: dibenzyldiamine, aminoadamantane, tricyclodecane-diamine, calcium tungstate, zirconium oxide, silica, silicone oil
TotalFill BC^™^	Zirconium oxide, calcium silicates, calcium phosphate monobasic, calcium hydroxide, filler and thickening agents

**Table 2 materials-14-06670-t002:** Mean values and SD of volume (in mm^3^) of root canal filling in the two experimental groups before retreatment.

Material	Mean	N	Std. Deviation	Minimum	Maximum	Median	Std. Error of Mean
AH Plus	15.30150	20	3.496143	10.200	21.700	15.01000	0.781761
TotalFillBC	14.08000	20	8.724343	4.680	35.310	11.86000	1.950822
Total	14.69075	40	6.589282	4.680	35.310	14.18500	1.041857

**Table 3 materials-14-06670-t003:** Mean values and SD of the volume (in mm^3^) of root canal filling in the two experimental groups and root canal remnants after root canal retreatment were determined by micro-CT analysis.

Material	Mean	N	Std. Deviation	Minimum	Maximum	Median	Std.Error of Mean
AH Plus	6.96450	20	2.703737	3.460	13.200	6.52500	0.604574
TotalFillBC	4.01350	20	3.125439	0.010	10.680	3.62000	0.698869
Total	5.48900	40	3.248577	0.010	13.200	5.30000	0.513645

## Data Availability

Data available from the author.

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
