# Peer review of "Efficacy of Reciprocating Instruments in the Removal of Bioceramic and Epoxy Resin-Based Sealers: Micro-CT Analysis"

_materials, 2021, doi:10.3390/ma14216670_

Round 1

Reviewer 1 Report

The authors aimed to assess the efficacy of reciprocating instruments in the removal of a calcium-silicate based sealer in an ex vivo model. The research question is relevant and the topic is under the scope of the journal. The manuscript is written in an easy way to follow. However, only one method was used for assessment (Micro-CT analysis) and the study model used is very distant to the clinical scenario, which deeply compromises the external validity of the results obtained. In this reviewer opinion, this is a simple model to do a very initial comparison between sealers, reasonable from an academic point of view, but not with the level of rigor necessary for a journal with this impact factor.

Introduction:

Authors show a bias in their reasoning, choosing studies with a higher difference in the outcome between initial endodontic treatment and orthrograde retreatment. The best evidence shous that the difference in the expected outcomes are not at that level (doi:10.1111/j.1365-2591.2011.01872.x – Ng et al 2011 IEJ. A prospective study of the factors affecting outcomes of nonsurgical root canal treatment: part 1: periapical health)

Authors also rely on in vitro references to support the reaction between calcium-silicate based sealers and tissular fluids to form hydroxyapatite and stimulate biomineralization, when in vivo studies more appropriately demonstrate those reaction

https://doi.org/10.3390/biomedicines9010024

Materials and methods:

The options for this section are very questionable. Initial treatment and retreatment were performed with the same type, size and geometry instrument. Clinical scenario is very different, professional have to approach root retreatment with knowing the instruments previously used and, therefore, removal of the material with be much more challenging, as well as the probability of left remnants behind will be higher.

Results:

A whisker and plot graphic with indications of statisticall significant differences between groups would be more gripping for readers. Information about the time used in the instrumentation of each group would also be interesting.

Discussion:

These results need at least to be compared with continuos rotation rotary instruments or with different irrigation regimens.

Reviewer 2 Report

The final conclusion that bioceramic is easier to remove than epoxy based needs a more convincing argument and I feel that the presented results (table 2) are lacking in detail.

Rather than presenting absolute volume measurements, consider presenting the mean residual filler percentage remaining (see Table 1 from https://www.ncbi.nlm.nih.gov/pmc/articles/PMC6249936/  ,  https://www.ncbi.nlm.nih.gov/pmc/articles/PMC6529792/  , or 
https://doi.org/10.1007/s00784-019-02956-3  )

The variance between initial & retreatment is visible however it would be useful to compare individual samples - did all 20 of the AH Plus samples experience a ~66% reduction in filling material? Or could it be that a few teeth were ideal and had complete removal, skewing the results? The large Std Dev for TotalFillBC treatment is suspect for both initial & retreatment measurements.
Perhaps a scatter plot (x-axis being tooth number, y-axis as volume of filling) could also make for a stronger argument.

If you do choose to keep Table 2 in some form, it looks like there has been a formatting error as there are multiple blank lines for each of the four groups (2/1/3/1 blanks respectively).

Given that other publications show minimal variation in retreatment between bioceramic & epoxy resin sealers (https://doi.org/10.1007/s00784-019-02956-3 as an example) can you please comment on your findings?
Could it be that the results obtained only apply for the tested bioceramic/epoxy resin comparison?

Reviewer 3 Report

Dear Authors,

The study written by Rajda et al, approaches a very interesting topic in dentistry and especially in endodontics.

Reciprocating instruments are contemporary considered one of the most efficient rotary solutions as they have compensated many disadvantages of the previous generations of rotary preparation techniques.

Retreatments are becoming more frequent and that's why I consider that improving clinical protocols in this field is essential.

I hope that my remarks will help the authors to improve the quality of the paper.

Line 79 - Please explain why you removed extraneous tissue and calculus from extracted teeth for a better understanding of the method.

Also please mention what type of curettes did you use and if you used scalers as well. Please also include the name of the manufacturer of the instruments.

Line 110 - Please clarify if you used manual files prior the rotary ones for retreatment.

Lines 237 - The conclusions part needs to be restructured and more elaborated in order to avoid sending a misleading message to the readers.

With the hope that my suggestions will be useful, I send you my best regards!

Round 2

Reviewer 1 Report

The authors performed the asked corrections and the manuscript improved.

Author Response

On behalf of all authors, I would like to thank the reviewers for the thorough review of the manuscript ‘Microtensile Bond Strength of Fiber Reinforced and Particulate
Filler Composite to Coronal and Pulp Chamber Floor Dentin’.

Reviewer 2 Report

Thank you for considering the previous comments & recommendations. 

In response to Reply 4, if you are reporting in absolute volume I feel that there should be a total volume value present in the paper for each material type. Reporting that there is 4.01/6.96mm^3 remaining (lines 181-182) but only showing the volume decrease in Table 2 is a little confusing. I suggest including an additional table (prior to Table 2) showing the same columns for the pre-treatment samples.

Some further minor suggestions:

  • Line 66 - space between showed and potential 
  • Lines 263-269 contain (almost) the same sentence twice
  • Table 2 (and Table 1.5 if you include it)
    • Rename mat column to Material or Group 
    • Right align the Mean column
    • Add leading 0 and right align Maximum column (i.e. .34 --> 0.34)
  • Adjust captions in Fig 2 & 3 to indicate that numbered/labeled points are indicative of samples with outlier values (is that correct?)

Reviewer 3 Report

Dear authors,

Thank you very much for the updated version.

Line 110-From my own experience the main access has to be performed using manual files. Please clarify that.

In the same time I consider that the English language needs improvement.

Please receive my best regards!
